# Estimation of Threshold Rainfall in Ungauged Areas Using Machine Learning

Kyung-Su Chu, Cheong-Hyeon Oh, Jung-Ryel Choi and Byung-Sik Kim *

Department of Urban and Environmental and Disaster Management, Graduate School of Disaster Prevention, Kangwon National University, Samcheok 25913, Korea; chu_93@kangwon.ac.kr (K.-S.C.); och@kangwon.ac.kr (C.-H.O.); lovekurt82@gmail.com (J.-R.C.)
* Correspondence: hydrokbs@kangwon.ac.kr; Tel.: +82-33-570-6819

**Abstract:** In recent years, Korea has seen abnormal changes in precipitation and temperature driven by climate change. These changes highlight the increased risks of climate disasters and rainfall damage. Even with weather forecasts providing quantitative rainfall estimates, it is still difficult to estimate the damage caused by rainfall. Damaged by rainfalls differently for inch watershed, but there is a limit to the analysis coherent to the characteristic factors of the inch watershed. It is time-consuming to analyze rainfall and runoff using hydrological models every time it rains. Therefore, in fact, many analyses rely on simple rainfall data, and in coastal basins, hydrological analysis and physical model analysis are often difficult. To address the issue in this study, watershed characteristic factors such as drainage area ($A$), mean drainage elevation ($H$), mean drainage slope ($S$), drainage density ($D$), runoff curve number ($CN$), watershed parameter ($L_p$), and form factor ($R_s$) etc. and hydrologic factors were collected and calculated as independent variables, and the threshold rainfall calculated by the Ministry of Land, Infrastructure and Transport (MOLIT) was calculated as a dependent variable and used in the machine learning technique. As for machine learning techniques, this study uses the support vector machine method (SVM), the random forest method, and eXtreme Gradient Boosting (XGBoost). As a result, XGBoost showed good results in performance evaluation with RMSE 20, MAE 14, and RMSLE 0.28, and the threshold rainfall of the ungauged watersheds was calculated using the XGBoost technique and verified through past rainfall events and damage cases. As a result of the verification, it was confirmed that there were cases of damage in the basin where the threshold rainfall was low. If the application results of this study are used, it is judged that it is possible to accurately predict flooding-induced rainfall by calculating the threshold rainfall in the ungauged watersheds where rainfall-outflow analysis is difficult, and through this result, it is possible to prepare for areas vulnerable to flooding.

**Keywords:** machine learning; random forest; regression analysis; support vector machine; threshold rainfall; threshold runoff; XGBoost

## 1. Introduction

Climate change has increased rainfall in Korea, resulting in various natural disasters that cause rapidly increasing social and economic loss [1]. However, Korean weather forecasts only provide rainfall information in absolute terms, and the same heavy rain warnings and special reports apply to all areas in Korea, which means a failure to reflect regional differences. For this reason, even with accurate forecasts, the forecast system fails to provide specific information on how different areas are affected and damaged by weather events. Forecasts focused on physical aspects of weather events do not provide sufficient information on how people's properties and safety are affected by them.

It is for this reason that the World Meteorological Organization (WMO) emphasizes the need for 'impact forecasts' that consider the socioeconomic effects that may be caused by weather events [2]. In Korea, different organizations provide different definitions of

impact forecasting. However, they can be summarized as follows: forecast that scientifically estimates the socioeconomic impact of weather at different times and places and delivers the estimates along with detailed weather information [3,4]. Outside of Korea, the WMO defines impact forecast as forecast that provides information on expected risks along with weather forecasts when disaster-causing high impact weather is expected. According to the Met Office of the United Kingdom, it is defined as a forecast that estimates the socioeconomic impact of a climate disaster at the time and place of its occurrence by considering meteorological disasters, level of exposure to disasters, and regional vulnerabilities. The National Weather Service of the United States defines it as a service aimed at providing the people with information on the social, economic, and environmental impact of weather, hydrological, and climate events [5]. Leading countries in the field of meteorology already provide information on socioeconomic impact of weather events along with high-resolution weather information. In the United Kingdom, the Flood Forecasting Centre (FFC) provides Flood Guidance Statements (FGS) that assess the risks of all flood types over five days and publish the findings daily [6,7]. The FFC uses the information to publish a table of flood risks which divides flood impact into four stages.

Impact forecasting requires threshold rainfall. Threshold rainfall means the rainfall amount that causes inundation. Accurate impact assessment requires calculation of the precise inundation-causing rainfall in each area. However, in Korea, research on threshold rainfall has been lacking. Most researchers use simplified analysis methods rather than refined hydraulic and hydrological analyses. Hydrological analyses of coastal areas are too complex to conduct properly.

As for previous literature on threshold rainfall calculation, ref [8] developed a flash food monitoring and prediction (F2MAP) model to calculate the flash flood-threshold runoff from rainfall. Ref [9] analyzed the relationship between flash flood index and runoff number characteristics to develop an equation between the two. Ref [10] proposed a threshold runoff calculation method using the flash flood guidance (FFG) model, which is more suitable for Korea rather than those used in the United States. The researchers presented the method as a way to acquire basic data for a flash flood forecast system. Ref [11] analyzed runoff in Jeju using the SWAT-K model that combines DEM, landcover, soil map methods, and developed a threshold runoff simulation method (TRSM) specifically for the island. Ref [12] used ArcGIS and HEC-GEOHMS to divide the Nakdonggang River watershed into 2268 sectors, drew rainfall-peak flow curves for different initial loss scenarios and antecedent moisture conditions, and calculated the threshold rainfalls. Ref [13] estimated threshold rainfalls for different durations using events with damage caused by past rainfall in urban areas and others without such damage. Ref [14] stressed the need for impact forecast and estimated threshold rainfalls using the SWMM model. Ref [7] also linked the grid base inundation analysis model (GIAM) for grid-based inundation analysis. Using the Huff distribution [14], the researcher converted the data into time-series rainfall data to simulate inundation depths, and inversely estimated the threshold rainfall based on the inundation depths. Ref [15] collected data on rainfall and typhoon damage over the last five years where inundation was caused, analyzed the relationship between rainfall and the damages, and developed an equation for threshold rainfall ($y = ax^b$). As can be seen from the literature cited above, Korean studies on threshold rainfall mostly used hydrological models. Few researchers studied threshold rainfall by considering hydrological characteristics.

More recently, a number of researchers used machine learning to improve the accuracy of threshold rainfall analysis [16–18]. Additionally growing is the body of literature that study rainfall-runoff, rainfall damage, and flood estimation with machine learning and deep learning rather than hydrological models [19–23]. However, few studies were identified in Korea that used machine learning to calculate threshold rainfall. Ref [19] sought to predict river water levels using observation data and deep learning algorithms. To that end, the researchers used tensor flow to predict water levels at the Okcheon Observatory location along the upper stream sectionof the Daecheong Dam within the Geumgang

watershed and used TensorFlow to develop a multiple regression model and a long short-term memory (LSTM) artificial neural network model. Ref [24] used three machine learning techniques (support vector machine, decision-making tree, and random forest) to develop a function for predicting rainfall damage in the Seoul Metropolitan Area (SMA) and found that support vector machine analysis using meteorological observation data from two days before yields the highest prediction performance. Ref [25] used the machine learning method on Gyeonggi-do, the province that suffers the worst rainfall damage each year. Choi used the data on rainfall damage of public facilities from the 2006–2015 Disaster Yearbooks published by the Ministry of the Interior and Safety (MIST) as the dependent variable. Ref [26] used machine learning methods such as ESN and DeepESN to predict rainfall using rainfall, pressure, and humidity from 2004 to 2014 as mediating variables. The correlation factors calculated using DeepESN yielded better results. Ref [27] performed hydrological rainfall adjustment using Light GBM and XGBoost. They found clear adjustment effects across all rainfall events after Light GBM and XGBoost learning, despite the fact that rainfall is adjusted 5 to 20 mm less.

Much of the literature cited above only used a single hydrological model. However, in this study, two models were coupled and used to calculate the marginal rainfall [8] and it is considered an advantage of this paper to apply the results to machine learning. Figure 1 shows the flow chart of this study. It was analyzed through machine learning using threshold rainfall and topographic factors of standard watershed units. The threshold rainfall was used as a dependent variable, and the topographic factor of the standard watershed unit was designated as an independent variable. In addition, the model with the smallest error was selected using error performance analysis to calculate the threshold rainfall in the ungauged basin where hydrological analysis was difficult. The ungauged basin means a coastal area where it is difficult to calculate the threshold rainfall.

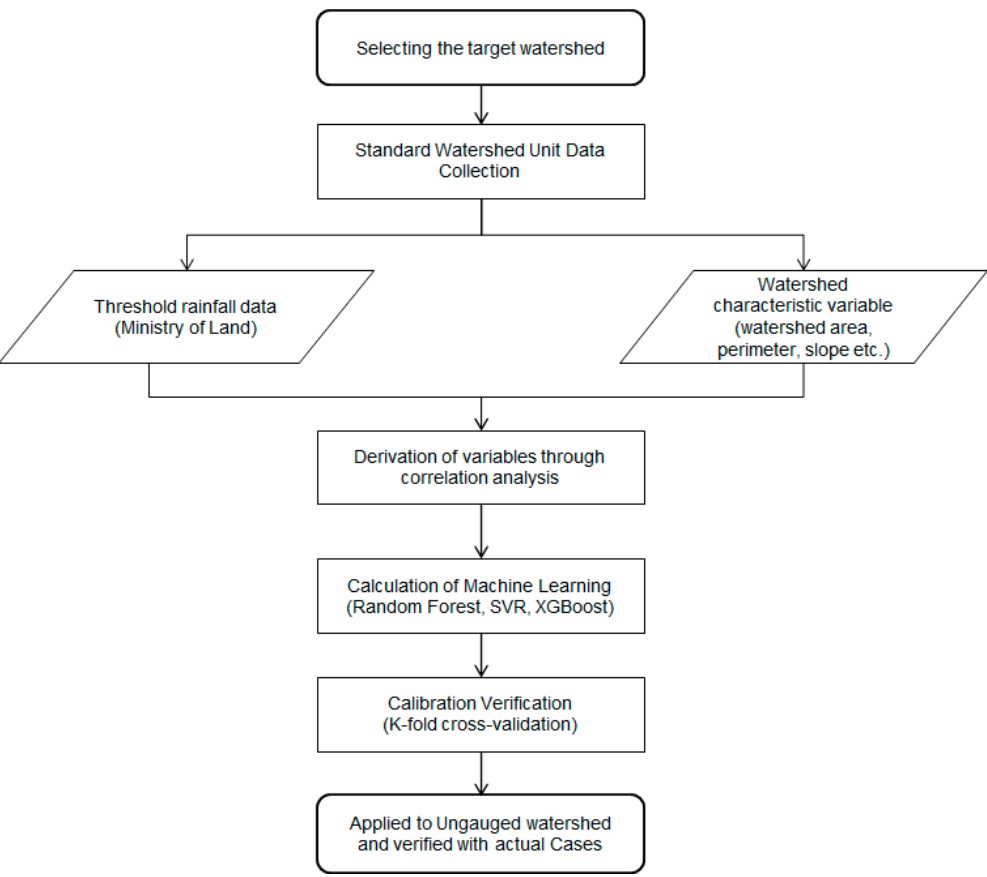

**Figure 1.** Flowchart of study.

## 2. Theoretical Background

### 2.1. Definition of Threshold Rainfall

In this study, threshold rainfall is calculated using the method used by the MOLIT in 2007, according to which threshold rainfall means the three-hour rainfall causing inundation depth at which the flow overflows the river embankment [8]. Threshold rainfall can be calculated by determining the rainfall of the rainfall-runoff curve corresponding to the threshold rainfall. In general, the runoff calculation equation for a rainfall-runoff model can be expressed as follows [15,28].

$$R_t = R_i + R_p \tag{1}$$

where $R_t$ is the total runoff, $R_i$ is the runoff at the impermeable layer, and $R_p$ is the flow at the permeable area.

$$R_t = \text{FFG} \times I + f(\text{FFG}) \times (I - 1) \tag{2}$$

In a rainfall-runoff model, rainfall and soil moisture constitute the inputs. However, the opposite is true with the flash flood threshold; calculation of flash flood threshold requires current soil moisture and required flow as inputs. As such, the equation on upper stream water and small-sized rivers is converted for FFG using the repetitive calculation method, as shown in Figure 2, to calculate the rainfall that causes threshold runoff. The FFG is the rainfall corresponding to the threshold runoff in the relationship of the rainfall and runoff curve. If there is no impervious area, the relationship between $R$ and FFG can be expressed in Figure 2 and Equation (2). $R$ means the threshold runoff (mm), FFG means the flash flow guidance (mm), and $f()$ means the fall-runoff process. Moreover, $I$ means rainfall intensity.

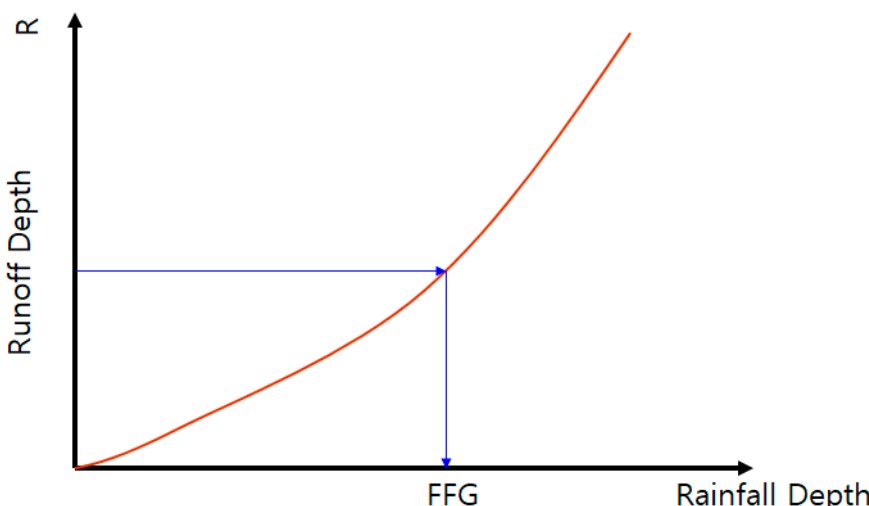

**Figure 2.** Concept of flash flood concept.

### 2.2. Machine Learning Method

Machine learning is an area of artificial intelligence where numerical models, algorithms, and programs are used to have a machine learn from given data as humans do, and new information is derived, or decisions are made based on what it learns [29]. In other words, machine learning means a system that uses accumulated empirical data to build models and improve performance. The amount of data matters in machine learning, and higher-quality data leads to higher-performing results. As for machine learning methods, this study used random forest, support vector machine, and XGBoost.

(1) Random Forest

The random forest method uses bootstraps to create several samples and applies them to a decision tree model to compile the results [30]. A decision-making trees produces estimates by creating and learning one-time training data from a given dataset. On the other

hand, a random forest creates multiple training data from a given data set and creates and combines multiple decision-making threes for improved prediction [31]. The observations not used by individual decision-making trees are out-of-bagging (OOB) data and are used for estimating prediction probability and identifying variables. The prediction probability of OOB observations for each observation $k$ within the $x_i$ category (0 or 10) [32].

This study used Python and the random forest method to calculate threshold rainfalls. Figure 3 shows the conceptual diagram of a random forest.

$$\hat{p}_k(x_i) = \frac{\sum_{j \in OOB_i} I\hat{y}(x_i, t_j) = k}{|OOB_i|} \text{ , for } k = 0, 1 \tag{3}$$

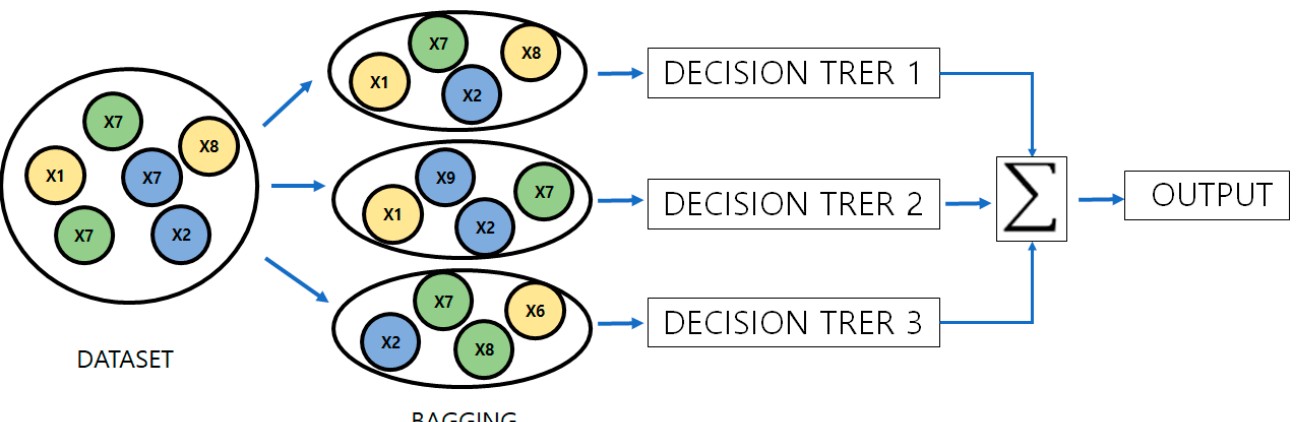

**Figure 3.** Concept of Random Forest.

(2) Support Vector Machine

Support vector machine (SVM) is a supervised learning algorithm used for both linear and non-linear classification issues. The purpose of SVM algorithms is to determine the lines or boundaries dividing an n-dimensional space into separate groups, so that they can be classified as their proper categories when new data are given. There may be multiple lines or boundaries for dividing an n-dimensional space into classes. However, the optimal boundary should be identified to determine categories. This optimal determination boundary is called the hyperplane. A support vector is the vector closest to the hyperplane and affects its position. The support vector machine is an algorithm that determines the optimal hyperplane that maximizes the margin, which means the distance between different data points.

The support vector regression (SVR) model has a small number of support vectors, and thus is known to be less sensitive to outliers. Ref [33] developed support vector regression that adopts a $\varepsilon$-insensitive loss function into the support vector machine. Support vector regression is estimated using the function shown in Equation (4) [32].

$$f(x) = \omega^t x + b \tag{4}$$

Equation (5) shows the constraints for calculating the optimal hyperplane function while calculating the error that minimizes Equation (4). $\frac{1}{2}\|\omega\|^2$ describes the degree of flattening of the function. If the data cannot be completely linearly separated, a slack variable $\xi(i = 1, \ldots I)$ is introduced to process it. $\xi$ means the distance between the margin and the data outside the boundary between the margins. The main superparameters of the support vector regression are $C$ (cost) and $\gamma$, and $C$ adjusts the complexity of the estimation model and the degree of error tolerance. An increase in $C$ means imposing a high penalty on errors within the margin. $\epsilon$ is not considered in the calculation process if the error is less than $\epsilon$ due to the maximum deviation between the actual value and the estimated value.

In this study, SVM was converted into SVR to predict arbitrary real values and used, and a Gaussian kernel (RBF) known for its excellent performance was applied. Figure 4 shows the conceptual diagram of the support vector machine.

$$\text{Minimuze}: \frac{1}{2} \parallel w \parallel^2 + C \sum_{i=1}^{I} (\xi_i + \xi_i^*)$$
$$\text{Subject to}: y_i - wx_i - b \le \epsilon + \xi_i$$
$$wx_i + b - y_i \le \epsilon + \xi_i^*$$
$$\xi_i \xi_i^* \ge 0$$

(5)

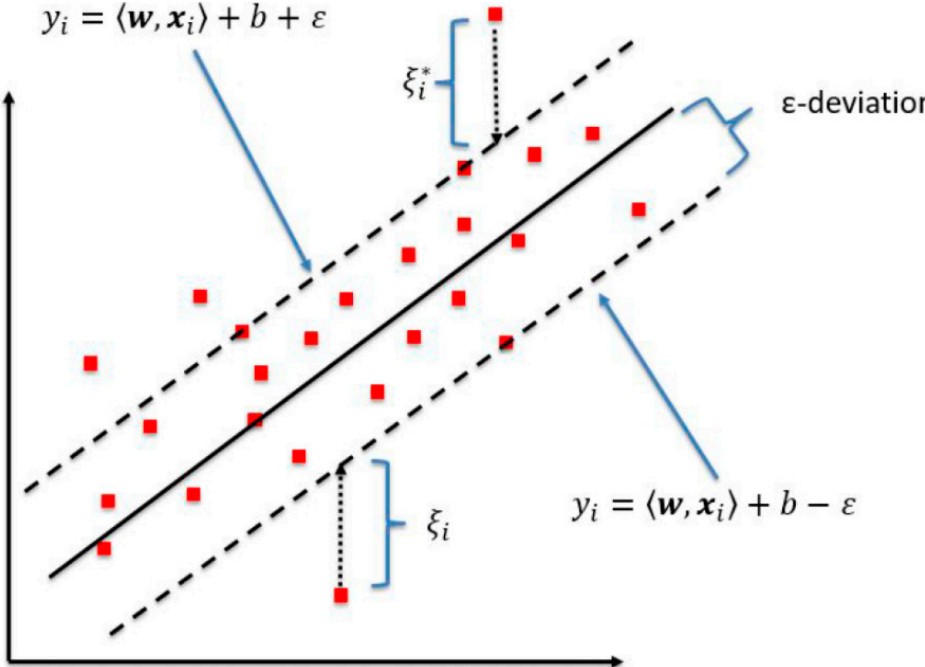

**Figure 4.** Concept of Support Vector Machine [34].

(3) eXtreme Gradient Boosting

Similarly to random forests, XGBoost is an ensemble algorithm that addresses the errors of multiple decision trees [35]. It offers improved prediction performance over gradient boosting machine (GBM) through distribution and parallel processing. In general, it is ten times faster than GBM. The efficiency and scalability of this method has been validated in multiple previous studies [36,37]. This boosting method lowers errors by grouping multiple classification and regression trees (CARTs).

$$\hat{y}_i = \sum_{k=1}^{k} f_k(x_i), f_k \in F$$

(6)

Equation (6) shows an ensemble model of trees, where *K* is the number of trees and *F* represents the set of CARTs. $f_k$ corresponds to the weight of each independent tree and leaf. The scores of the leaves are summed up and compared for final prediction.

$$Obj = \sum_{i}^{n} = l(y_i, \hat{y}_i) + \sum_{k=1}^{K} \Omega(f_k)$$

(7)

Equation (7) represents an XGBoost model. The first $l(y_i, \hat{y}_i)$ is a loss function that represents the difference between a prediction and an actual observation. The second $\Omega(f_k)$ is the normalization term that controls the complexity of the model to prevent overfitting.

*2.3. Performance Assessment Using K-Fold Cross Validation*

This study uses MAE, RMSE, and RMSLE to compare the performance of different models. Most studies use the above three indicators a lot for data comparison [38–40]. They

are widely used to objectively assess the accuracy of a regression equation by analyzing differences between observations and estimates. MAE and RMSE are statistical indicators for confirming the degree of errors included in an estimate calculated using an equation, when compared with an observation. A value closer to 0 represents better fit. RMSLE represents the average ratio of observations to predictions.

$$\text{MAE} = \frac{1}{N} \sum_{i=1}^{N} |y_i - \hat{y}_i| \tag{8}$$

$$\text{RMSE} = \sqrt{\frac{1}{n} \sum_{i=1}^{n} (y_i - \hat{y}_i)^2} \tag{9}$$

$$\text{RMSLE} = \sqrt{\frac{1}{n} \sum_{i=1}^{n} (\log(y_i + 1) - \log(\hat{y}_i + 1))^2} \tag{10}$$

Ideally, these errors need to be tested by applying them to actual ungauged watersheds. However, due to data and time constraint, the prediction models were validated using five-fold cross validation. K-fold cross validation is a model assessment method that uses a part of the overall data as a validation set. It ensures that all data are used as dataset at least once. Figure 5 shows dividing the data into five datasets and validating the models with a different dataset each time. An average cross-validation uses five datasets. This study selects the optimal parameters following cross validation to calculate threshold rainfalls.

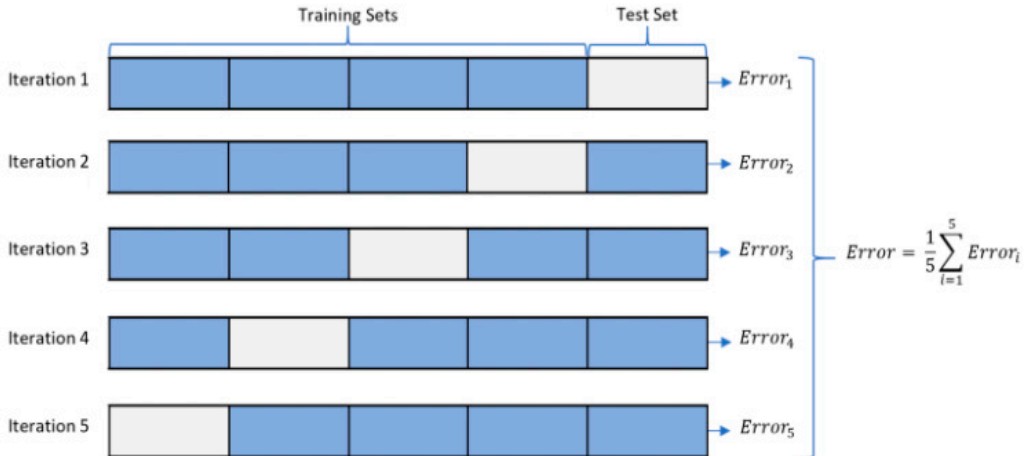

**Figure 5.** Concept of cross vaildation.

## 3. Selection of Target Watersheds and Variables

### 3.1. Selection of Target Watersheds

This study chose the Han River watershed as its target, as the area includes the highest number of standard watersheds according to the water resource unit map. 290 of Korea's 850 standard watersheds are included in the Han River watershed. 237 of the 290 watersheds are inland, and the other 53 are coastal watersheds, as shown in Figure 6.

The (a) section of Figure 7 shows the learning watersheds; 80% of the learning watersheds were used for machine learning, and the other 20% were used for validation. High-performing models were selected with (a), and predictions were performed for the watersheds highlighted yellow in (b). The data set of the basin used for machine learning was randomly selected and proceeded.

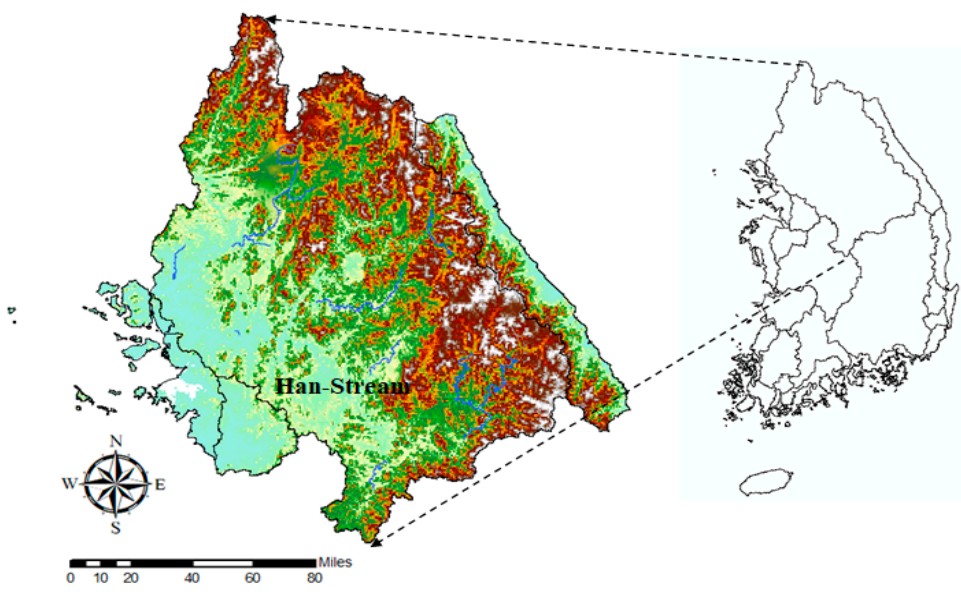

**Figure 6.** Study Areas.

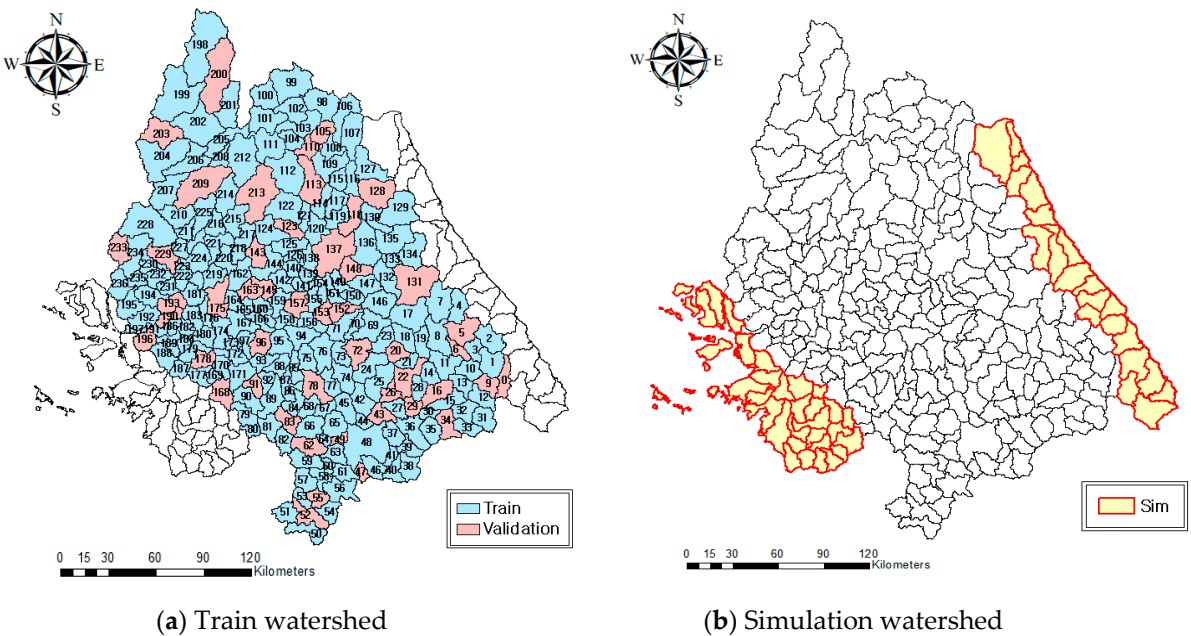

(**a**) Train watershed        (**b**) Simulation watershed

**Figure 7.** Training and Prediction Watersheds for Machine Learning.

*3.2. Dependent and Independent Variables*

This study used the threshold rainfalls calculated using the MOLIT method [8,15,41] as dependent variables. Figure 8 shows the calculated threshold rainfalls on the map.

Characteristic factors of the watersheds were used as independent variables. The analysis only considered topographical factors and hydrological factors. The watershed characteristic factors used in were collected from the Water Resources Management Information System (www.wamis.go.kr, accessed on 31 December 2011) and the geographic information system (GIS). Data on 15 characteristic factors were collected, including: drainage area (km$^2$), mean drainage elevation (m), mean drainage slope (%), highest drainage elevation (m), drainage density, runoff curve number, river length (km), drainage perimeter (km), form factor, circularity ratio, stream frequency, channel maintenance constant, relative relief, number of reliefs, and river length ratio.

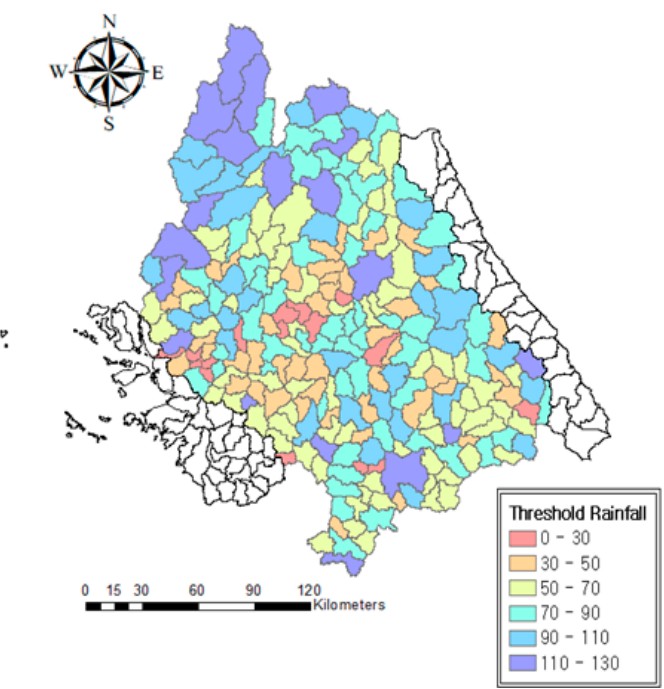

**Figure 8.** Standard watershed Threshold rainfall.

The drain area refers to the area on the plane of the basin and refers to the plane area within the closed curve, which is usually made up of a watershed. The basin average elevation is calculated by arithmetically averaging the elevation values corresponding to each cell of the DEM (Digital Elevation Model). The mean drain slope is calculated by arithmetically averaging the slope corresponding to each cell of the DEM in degrees. Highest drain evaluation means the highest elevation in the basin, and drain density means the length of rivers per unit area. It means that the degree of outflow of the basin is quantified by the Soil Conservation Service (SCS) using the runoff curve number land use and soil map. River length is the total length of all rivers in a given drainage basin. The drain perimeter is defined as the length measured along the boundary of the watershed of a given order projected on the horizontal plane of the map, and the form factor is defined as the ratio of the main river length of the watershed to the diameter of the circle having the same area as the watershed area. The circularity ratio is a dimensionless parameter defined as the ratio of the basin area to the area of a circle with the same length as the basin circumference. Stream frequency is defined as the ratio of river water in the basin to the basin area, and the channel maintenance constant is the reciprocal of the aqueous density. Relative relief is defined as the ratio of watershed undulations to watershed circumference, number of reliefs is defined as the product of watershed undulations and water density, and river length ratio is defined as the ratio of river length $w$ to average river length $w - 1$. Table 1 shows a summary of the watershed characteristics factor.

**Table 1.** Summary of independent variables.

|  | $A$ | $H$ | $S$ | $Em$ | $H$ | $CN$ | $L_w$ | $L_p$ | $R_s$ | $R_c$ | $Cf$ | $C$ | $R_p$ | $R_n$ | $RL$ |
|---|---|---|---|---|---|---|---|---|---|---|---|---|---|---|---|
| Count | 290 | 290 | 290 | 290 | 290 | 290 | 290 | 290 | 290 | 290 | 290 | 290 | 290 | 290 | 290 |
| mean | 144.6 | 324.0 | 35.4 | 253.3 | 1.7 | 58.7 | 12.9 | 67.3 | 1.0 | 0.4 | 2.4 | 0.7 | 13.3 | 1376 | 1.8 |
| Max | 571.6 | 930.3 | 65.1 | 302.7 | 4.0 | 87.9 | 63.3 | 262.3 | 3.6 | 0.7 | 12.6 | 9.5 | 36.8 | 3921.4 | 4.4 |
| min | 39.0 | 4.9 | 4.0 | 103.7 | 0.1 | 33.7 | 0.0 | 32.7 | 0.0 | 0.0 | 0.1 | 0.3 | 0.9 | 32.8 | 0.7 |

A correlation analysis was performed to select statistically correlated independent variables, as independent variables not correlated to dependent variables may lower the

prediction performance. The correlation analysis was performed as shown in Figure 9. Given the fact that the threshold runoffs required for calculating threshold rainfalls were calculated from peak flood volumes and overflowing runoffs, the following variables were determined to be significantly correlated: drainage area, river length, drainage perimeter, relative relief, and river length ratio. Among those factors, river length, river length ratio, relative relief, and drainage perimeter were determined to be more highly correlated with the independent variables. The correlation coefficient was 0.65 for threshold rainfall and drainage area, 0.64 for drainage perimeter, and 0.31 for river length. As such, drainage area, drainage perimeter, and river length were finally selected as independent variables. In most data analyses, principal component analysis (PCA) should be used to reduce initial independent variables [42], but in this study, a principal component analysis was omitted because the amount of data for each independent variable was not large.

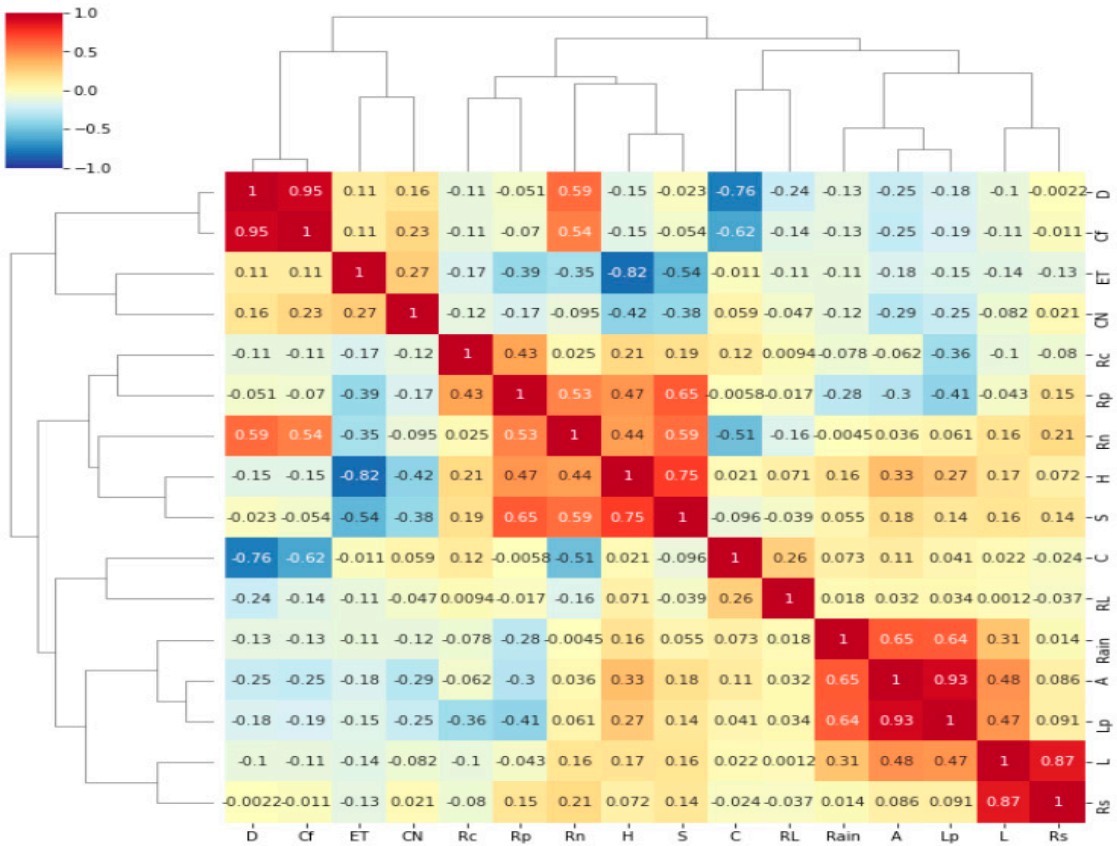

**Figure 9.** Variable correlation analysis.

## 4. Machine Learning Application and Results

This study used SVM, random forest, and XGBoost. Excel 2010 and Python ver. 3.6 were used to record and statistically analyze the collected data and generate graphs. This study also used the model packages provided by Python-based Scikit-learn.

Effective machine learning requires pre-processing of the data to be used. The independent variable data went through data scaling and missing values were removed. As data scales vary depending on the variable, the data were standardized to render them more suitable for machine learning. Independent variables were analyzed using RobustScaler, which is less affected by outliers. A higher accuracy can be expected by removing outliers. However, the small number of inputs in this study means possible overfitting. Therefore, this study addressed outliers through pre-processing rather than outlier removal.

### 4.1. Validation of Prediction Models

The optimal parameters for each model were selected through K-fold cross-validation. An error closer to 0 indicates a better result. Table 2 shows the MAE, RMSLE, and RMSLE values of each of the five datasets created by dividing the datasets through the k-fold cross-validation. All three performance assessments found that XGBoost produces the results closest to actual observations compared with the other models.

**Table 2.** Comparison of model performance evaluation.

| Model | | MAE | RMSE | RMSLE |
|---|---|---|---|---|
| Support Vector | Fold 1 | 15 | 19 | 0.26 |
| | Fold 2 | 23 | 38 | 0.4 |
| | Fold 3 | 19 | 26 | 0.29 |
| | Fold 4 | 21 | 26 | 0.47 |
| | Fold 5 | 28 | 38 | 0.34 |
| Random Forest | Fold 1 | 12 | 19 | 0.28 |
| | Fold 2 | 20 | 32 | 0.46 |
| | Fold 3 | 16 | 20 | 0.34 |
| | Fold 4 | 22 | 27 | 0.47 |
| | Fold 5 | 21 | 26 | 0.45 |
| XGBoost | Fold 1 | 14 | 20 | 0.28 |
| | Fold 2 | 20 | 33 | 0.38 |
| | Fold 3 | 16 | 20 | 0.29 |
| | Fold 4 | 21 | 27 | 0.46 |
| | Fold 5 | 25 | 37 | 0.35 |

Parameters were applied to increase the accuracy of machine learning, and n_estimator represents variables that adjust the number of trees to generate. max_depth means the number of tree depths. min_samples_split represents the minimum number of sample data to split nodes, and min_samples_leaf means the minimum number of sample data required for a leaf node. learning_rate means a parameter that, in machine learning and statistics, moves toward the min loss function and determines the size of each stage of repetition. The calculated parameter values area n_estimators: 100, learning_rate = 0.04, min_samples_leaf = 3, min_samples_split = 2, max_depth = 4.

Figure 10 compares the existing threshold rainfalls with those calculated using XG-Boost. Most threshold rainfalls are distributed between 40 and 60 mm and between 60 and 80 mm and are close to actual observations.

Figure 11 is a map representing threshold rainfall values. The watersheds with low threshold rainfalls in (a) are reflected in (b) as well.

### 4.2. Calculation of Threshold Rainfalls in Ungauged Watersheds

This study used XGBoost, which produced good results in error performance assessment, to calculate the threshold rainfalls of ungauged watersheds. Figure 12 shows the distribution of the threshold rainfalls calculated for the ungauged basins. The majority of watersheds show threshold rainfalls between 40 mm and 80 mm. Figure 13 is a map showing the threshold rainfalls of the ungauged basins other than the those in the inland areas.

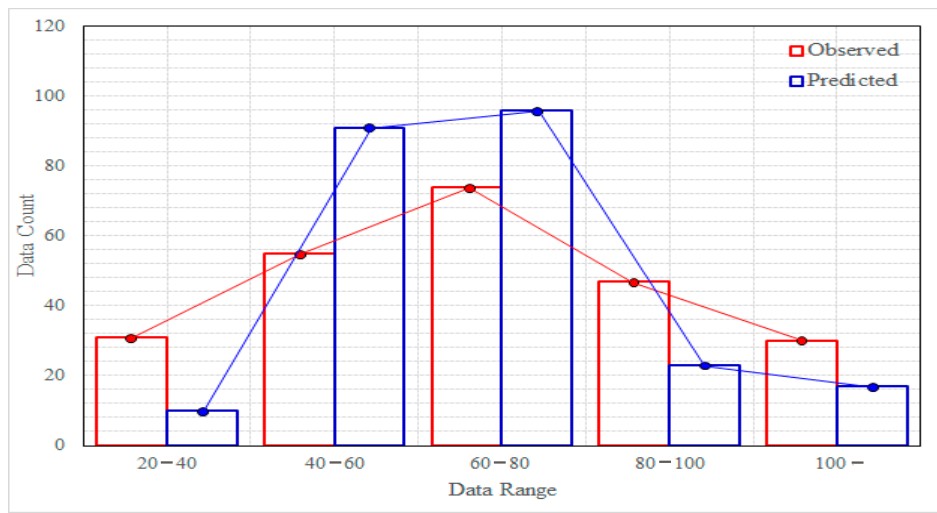

**Figure 10.** Threshold rainfall distribution.

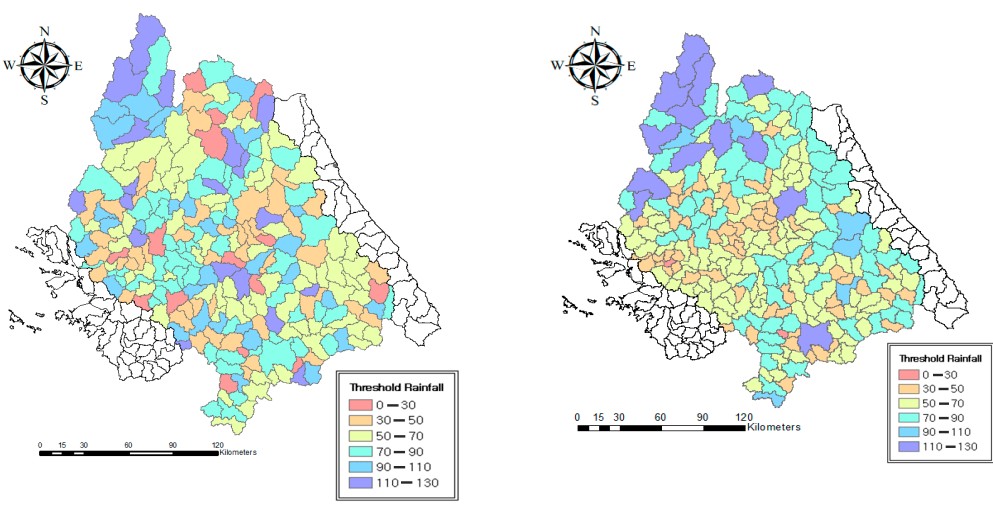

(**a**) Estimated limit rainfall value         (**b**) Threshold rainfall value using XGBoost

**Figure 11.** Comparison of training data against actual values.

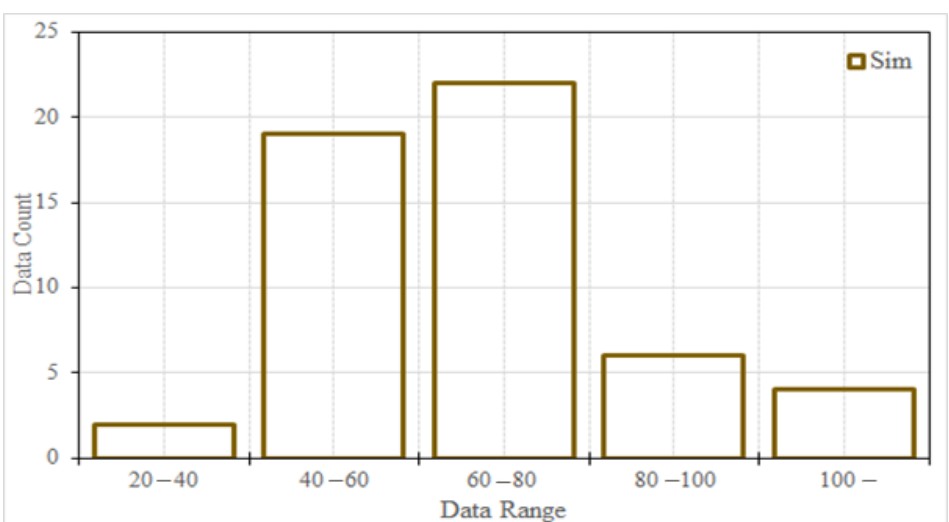

**Figure 12.** Ungauged watershed threshold rainfall distribution.

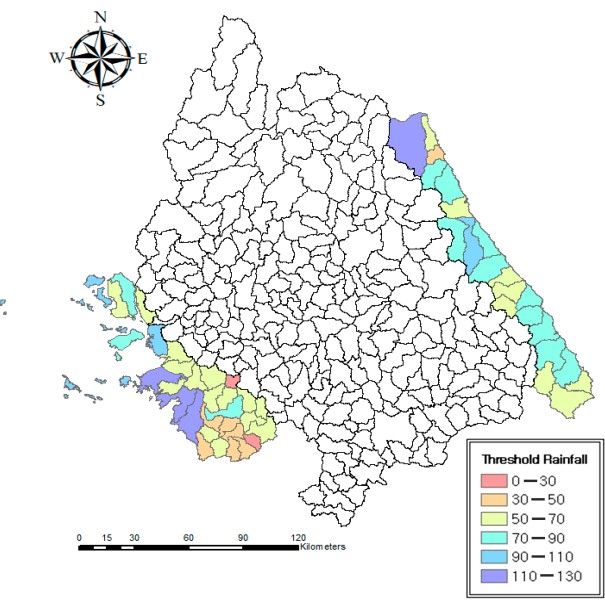

**Figure 13.** Threshold rainfall values for ungauged watersheds.

*4.3. Validation Using Real World Cases and Assessment*

As shown in Figure 14, based on the rainfall events in 2017, 2020, and 2021 of each affected watershed, among the ungauged watersheds outside the purple lines around the inland part of the Han River, Yongin, Cheonan, Samcheok, Gangneung, and Sokcho watersheds were found to be vulnerable against heavy rain. An application to actual rainfall events showed that damage was caused when the rainfall exceeds the specified rainfall in the legends. However, the researchers' ability to verify damages in other areas was restricted by the fact that damage was verified from news articles and social network posts.

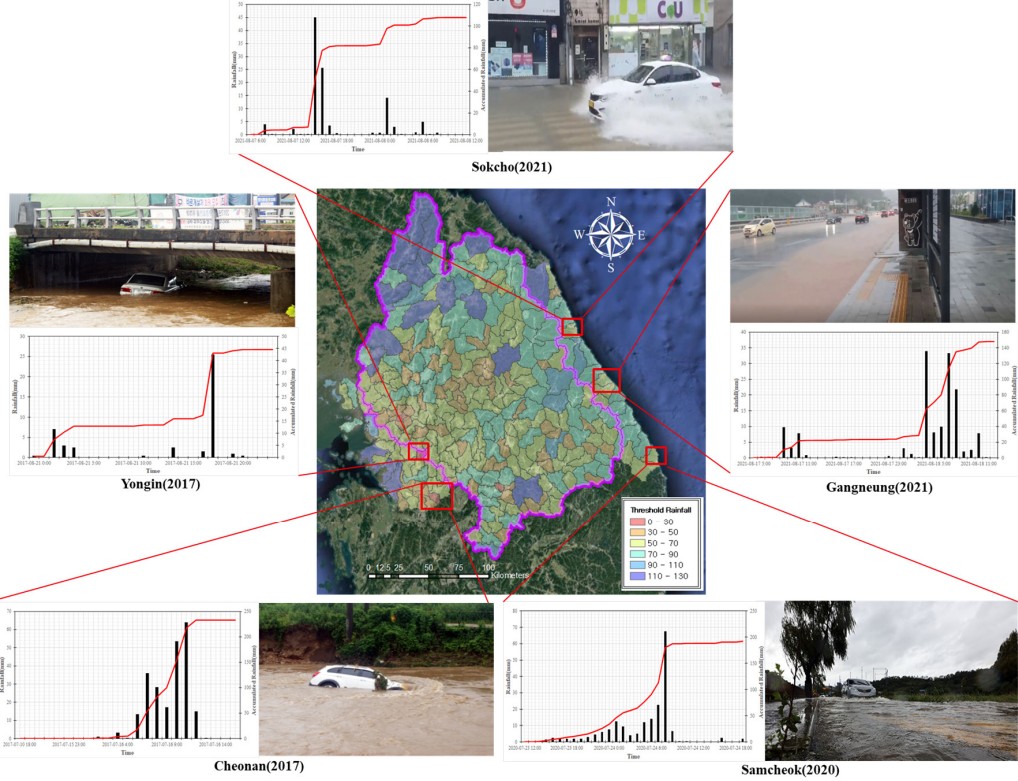

**Figure 14.** Threshold rainfall values for ungauged watersheds.

## 5. Conclusions

Damage caused by localized heavy rain continues to increase in Korea. However, research on inundation-causing threshold rainfall is still largely absent in the country. More researchers need to study technologies for predicting and responding to inundation in advance. This study can be summarized as follows.

The purpose of this study was to identify threshold rainfalls in areas not readily available for hydrological analysis, using the calculation method and characteristic factors used by the MOLIT. Three machine learning methods (SVM, random forest, and XGBoost) were compared in terms of accuracy using MAE, RMSE, and RMSLE, and XGBoost was selected as the best-performing method. Watershed characteristics, hydrological factors, and XGBoost were used to calculate the threshold rainfalls of the ungauged coastal watersheds. In this study, it is judged that what can reflect actual topographic and hydrological factors can be differentiated from other machine learning and marginal rainfall papers. In addition, distinct from conventional simple data, data using physical models were used in machine learning techniques, so high accuracy could be secured through a small number of data, and anyone could use it by using widely known machine learning techniques.

However, this study has its limitations. First, outliers were found while calculating the threshold runoffs of the hydrological models. More sophisticated hydrological models and more accurate data may be needed for analysis. In addition, threshold rainfall calculation based on the runoff-rainfall curve simply used polynomials. However, higher accuracy may be achieved by applying a machine learning method to threshold runoff and runoff-rainfall curve calculation.

This study compared the calculated threshold rainfalls with real world cases identified from news reports and social network posts, which was found to pose limitations to quantitative assessment.

The researcher plans to conduct a similar study nation-wide. Watershed data with more diverse hydrological models and outliers will improve the accuracy of the findings. Although not included in this study, quantitative validation using real world events will yield meaningful results. The current water forecast system provides only quantitative figures without considering the damage caused, which some regard as insufficient for supporting effective decision-making to prevent and prepare for damage caused by natural disasters. Threshold rainfall prediction suggested in this study may, if implemented on a continued basis, provide accurate information on rainfall damage in advance and help decisionmakers make better decisions on disaster control.

**Author Contributions:** K.-S.C. and C.-H.O., J.-R.C. carried out the survey of previous study and wrote the graph of the data. B.-S.K. suggested idea of study and contributed to the writing of the paper. All authors have read and agreed to the published version of the manuscript.

**Funding:** This work was funded by the Korea Meteorological Administration Research and Development Program under Grant KMI [2021—00312]. This work was financially supported by Ministry of the Interior and Safety as Human Resource Development Project in Disaster Management(C2001777-01-01).

**Data Availability Statement:** Not applicable.

**Conflicts of Interest:** The authors declare no conflict of interest.

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
