# Peer review of "Estimation of Threshold Rainfall in Ungauged Areas Using Machine Learning"

_water, doi:10.3390/w14060859_

Round 1

Reviewer 1 Report

This subject addressed is within the scope of the journal. However, the manuscript in the present version contains several problems. Appropriate revisions should be undertaken in order to justify recommendation for publication.

  1. No decomposition method is used for decomposition to capture data noise. why? How will this affect the results? More details should be furnished.
  2. It is mentioned that XGboost and SVM are used as main models. What are the advantages of adopting these particular method over others in this case? How will this affect the results? More details should be furnished. Why not tried hybrid models for comparison? For example,LSTM-ALO,DENFIS,GMDH,OP-ELM,LSSVM-GSA recently used in literature streamflow modeling. Should add these models recent literature and also explain why not adopted those advanced version? Authors should compare MARS or M5Tree model for comparison to validate their model results.
  3.     For readers to quickly catch your contribution, it would be better to highlight major difficulties and challenges, and your original achievements to overcome them, in a clearer way in abstract and introduction.
  4.     It is mentioned that korea is adopted as the case study. What are other feasible alternatives? What are the advantages of adopting this case study over others in this case? How will this affect the results? The authors should provide more details on this.
  5. There is a serious concern regarding the novelty of this work. What new has been proposed?
  6. Abstract needs to modify and to be revised to be quantitative. You can absorb readers' consideration by having some numerical results in this section.
  7. There are some occasional grammatical problems within the text. It may need the attention of someone fluent in English language to enhance the readability.

  8. 8. Since the some figures have low-resolution printing, the reviewer cannot recognize them clearly. Please revise them with high resolution.
  9. The discussion section in the present form is relatively weak and should be strengthened with more details and justifications.
  10. In conclusion section, limitations and recommendations of this research should be highlighted.
  11. The authors have to add the state-of-the art references in the manuscripts.
  12.    Some key parameters are not mentioned. The rationale on the choice of the set of parameters should be explained with more details. Have the authors experimented with other sets of values? What are the sensitivities of these parameters on the results?
  13. It is mentioned that three performance indexes were used. What are the advantages of adopting these indexes over others (CC, willimot index) in this case? How will this affect the results? More details should be furnished.

Author Response

Response to Reviewer 1 Comments

This subject addressed is within the scope of the journal. However, the manuscript in the present version contains several problems. Appropriate revisions should be undertaken in order to justify recommendation for publication

Response : Thank you so much for the detailed review.

Point 1: No decomposition method is used for decomposition to capture data noise. why? How will this affect the results? More details should be furnished.

Response 1: Thank you for your opinion. I derived independent variables through correlation analysis in this study to reduce data noise and responded to the outliers as much as possible using Python's robustscaler technique to remove the outliers of the derived variables. However, this study aims to generate data through actual topographic and river factors, so we adopted a method of reflecting noise to some extent and maximizing the model performance of machine learning. (Line 367-376)

Point 2 : It is mentioned that XGboost and SVM are used as main models. What are the advantages of adopting these particular method over others in this case? How will this affect the results? More details should be furnished. Why not tried hybrid models for comparison? For example,LSTM-ALO,DENFIS,GMDH,OP-ELM,LSSVM-GSA recently used in literature streamflow modeling. Should add these models recent literature and also explain why not adopted those advanced version? Authors should compare MARS or M5Tree model for comparison to validate their model results.

Response 2 : I know that the model mentioned by the reviewer shows excellent performance in time series prediction. If you give me related references, I will be able to write a reference that there is a good model for the research content.
In this study, a model that is widely used and easily used among multivariate regression models was used, and of course, as the reviewer said, there is a plan to analyze it through other models. I added this briefly to the conclusion. Thank you.

Point 3: For readers to quickly catch your contribution, it would be better to highlight major difficulties and challenges, and your original achievements to overcome them, in a clearer way in abstract and introduction.

Response 3: Other reviewers also mentioned abstract revision. I made a new abstract. Thank you for the review. (Line 11-43).

Point 4: It is mentioned that korea is adopted as the case study. What are other feasible alternatives? What are the advantages of adopting this case study over others in this case? How will this affect the results? The authors should provide more details on this.

Response 4 : As mentioned in the introduction, the differentiation between this study and other studies was not only calculated by combining existing physical models, but also considered actual topographic factors. Other studies conducted machine learning only with simple weather data, but this study analyzed machine learning with the results from the model simulation, so it is judged that the data size was small but could show high accuracy. Thank you for your specific opinion.

Point 5: There is a serious concern regarding the novelty of this work. What new has been proposed?

Response 5 : I think it's a similar question to the point mentioned earlier. To give you an additional answer, this study did not end up calculating the model through simple machine learning, and even reviewed the applicability through case verification of the unmetered basin. In particular, the coastal topography in Korea has a lot of river trunks, making it difficult to apply physical models. In this regard, machine learning techniques that reflect actual topographic factors will certainly be available abroad, and anyone can easily follow them using machine learning techniques that are not difficult.

Point 6 : Abstract needs to modify and to be revised to be quantitative. You can absorb readers' consideration by having some numerical results in this section.

Response 6 : I modified the abstract overall. Thank you. (Line 11-43).

Point 7 : There are some occasional grammatical problems within the text. It may need the attention of someone fluent in English language to enhance the readability.

Response 7 : I will revise the entire paper through foreign inspection when all the reviewer's revisions are reflected. Thank you.

Point 8: Since the some figures have low-resolution printing, the reviewer cannot recognize them clearly. Please revise them with high resolution.

Response 8 : The picture was captured and used directly. I modified Figures 3 and 4 to a high resolution.

Point 9: The discussion section in the present form is relatively weak and should be strengthened with more details and justifications.

Response 9 : Other reviewers gave the same opinion. Overall, the paper was revised.
Thank you.

Point 10 : The discussion section in the present form is relatively weak and should be strengthened with more details and justifications.

Response 10 : I added a new part of the conclusion.

Point 11 : The authors have to add the state-of-the art references in the manuscripts.

Response 11 : I added the references mentioned by the reviewers.

Point 12 : Some key parameters are not mentioned. The rationale on the choice of the set of parameters should be explained with more details. Have the authors experimented with other sets of values? What are the sensitivities of these parameters on the results?

Response 12 : The parameters collected in this study are variables provided by the state and those calculated through GIS. I added a detailed description of the variables to Line 325-342 and composed of the factors necessary to calculate the threshold rainfall.

Point 13 : It is mentioned that three performance indexes were used. What are the advantages of adopting these indexes over others (CC, willimot index) in this case? How will this affect the results? More details should be furnished.

Response 13 : I'm really sorry, but I don't know exactly what it's about. If you can afford it, tell me in more detail and I will proceed with the correction immediately.

Reviewer 2 Report

The purpose of this paper was capture threshold rainfall using Machine Learning techniques. The independent variable along with five independent variables are used by three machine learning models so as to derive the threshold rainfall while XGBoost was found to be the optimum model for calculating threshold value in ungauged watersheds.  This is an interesting study which in general presents a specific, easily identifiable advance in knowledge on calculating threshold value in ungauged watersheds. However, Principal component analysis (PCA) should be used   to reduce the 15 initially explanatory indicators while more details should be provided regarding the model validation using real cases. Finally, I believe that the following revisions must be taken into account in order to optimize the manuscript so as to be suitable for publication.

  1. Some quantitative results must be incorporated in the abstract.

  1. Lines 106-108: The purpose of this study should be clearly stated along with the means which are going to be used so as to achieve this purpose.

  1. Regarding the wide usage of the MAE, RMSE and RMSLE some reference could be used so as to further support their statement e.g. Ioannou, K.; Myronidis, D.; Lefakis, P.; Stathis, D. The use of artificial neural networks (ANNs) for the forecast of precipitation levels of lake Doirani (N. Greece). Fresenius Environ. Bull. 2010, 19, 1921–1927

  1. Line 256: It should be stated how the dataset was divided into learning and validation watersheds.

  1. Lines 281-290: Principal component analysis (PCA) should be used  to reduce the 15 initially explanatory indicators while the following citation can provide further details on this process Myronidis, D.; Ivanova, E. Generating regional models for estimating the peak flows and environmental flows magnitude for the Bulgarian-Greek Rhodope mountain range torrential watersheds. Water 2020, 12, 78

  1. Figure 10 could be omitted since all necessary information is provided in Table 2. 

  1. Line 321: It seems that something is missing at the beginning of the paragraph.

  1. Figure 11 could be omitted since all necessary information is provided in Table 2. 

  1. Lines 350-365: The model validation analysis is weak and more details are necessary regarding the model validation using real cases.

Author Response

Thank you!!

Reviewer 3 Report

Manuscript ID: water-1587665

Manuscript title: Estimation of Threshold Rainfall in Ungauged Areas using Machine Learning

Authors: KyungSu Choo, CheongHyeon Oh, JungRyel Choi, and Byungsik Kim

Summary:

            In this manuscript, the authors used three machine learning (ML) methods (Support vector machine method-SVM, random forest method, and eXtreme Gradient Boosting-XGBoost) to estimate the threshold rainfall in ungauged areas (coast areas of Korea). The performance of the considered three ML methods is estimated using root mean square error (RMSE), mean absolute error (MAE), and root mean square logarithmic error (RMSLE). The authors demonstrated that the XGBoost produced good results among all performance assessments. The manuscript is written clearly and concisely. However, there are some typo errors in the manuscript in some instances. The authors can consider the below-mentioned minor comments before accepting the manuscript to the Water journal.

Minor comments:

  1. Page 2, lines 76-77: “Lee also” the cited reference is not mentioned as per the journal format.
  2. Page 2, line 78: “Huff, 1967” Please change the citation format as per the journal style.
  3. Page 3, line 89: Please check that the text has a similar font size and font type.
  4. Page 3, line 91: “setion” or “section”?
  5. Page 3, line 105, page 12, 333, Page 13, line 348: A space between the units and their numerical value is needed. Please also check other places of the manuscript and make the necessary modifications.
  6. Page 4, Figure 1: A little more description of Figure 1 should be provided in its caption.
  7. The quality of the figures seems little poor. It is recommended to provide all the figures with a minimum resolution of 300 dpi or more.
  8. Page 4, line 133: Please describe the variables in Equation 2.
  9. Page 5, liens 138-139: What green dotted lines and solid red line in 2 represent. Please mention them in the manuscript.
  10. Page 5, line 159: “threes” or “trees”?
  11. Page 4, line 165: Please describe all Equation (3) variables in the manuscript.
  12. Page 6, line 175: Correct the typo error in line 175.
  13. Page 6, line 183-185: The sentence is unclear; please rewrite it.
  14. Page 6, lines 192-196: Please describe all Equation (5) variables in the manuscript.
  15. Page 6, Figure 4. A little more description about Figure 4 is recommended.
  16. Page 7, Equation (8) to (10): describe the parameters in Equations (8) to (10)
  17. Page 9, Table 1: Please detail all the listed parameters in the Table caption.
  18. Page 10, line 298: Correct the typo error in line 298.
  19. Page 8, line 256, Page 12, line 332, 335: Please check that the text has a similar font size and font type.
  20. Page 15, lines 381-382: The sentence is unclear; please rewrite it.

Author Response

Thank you!!

Round 2

Reviewer 1 Report

Authors revised properly.should accept now.

Author Response

Response to Reviewer 1 Comments

Authors revised properly.should accept now.

Response : Thank you so much for the detailed review!

Reviewer 2 Report

Reference N. 36 should be corrected. to:   Ioannou, K.; Myronidis, D.; Lefakis, P.; Stathis, D. The use of artificial neural networks(anns) for the forecast of precipitation levels of lake doirani(N. greece). Fresenius Environ. Bull. 2010, 19, 1921–1927.

Author Response

Response to Reviewer 2 Comments

Reference N. 36 should be corrected. to:   Ioannou, K.; Myronidis, D.; Lefakis, P.; Stathis, D. The use of artificial neural networks(anns) for the forecast of precipitation levels of lake doirani(N. greece). Fresenius Environ. Bull. 2010, 19, 1921–1927.

Response : I modified the reference No. 40. Thank you so much for the detailed review!